# Mortality during 6 years of follow-up in relation to visual impairment and eye disease: results from a population-based cohort study of people aged 50 years and above in Nakuru, Kenya

Hannah Kuper,[1,2] Wanjiku Mathenge,[3] David Macleod,[4] Allen Foster,[1,2] Michael Gichangi,[5] Hillary Rono,[1,6] Kevin Wing,[7] Helen Anne Weiss,[4] Andrew Bastawrous,[1] Matthew Burton[1]

► Prepublication history and additional material is published online only. To view please visit the journal online (http://dx.doi.org/10.1136/bmjopen-2019-029700).

For numbered affiliations see end of article.

**Correspondence to**
Professor Hannah Kuper;
hannah.kuper@lshtm.ac.uk

## ABSTRACT

**Objective** To estimate the association between (1) visual impairment (VI) and (2) eye disease and 6-year mortality risk within a cohort of elderly Kenyan people.

**Design, setting and participants** The baseline of the Nakuru Posterior Segment Eye Disease Study was formed from a population-based survey of 4318 participants aged ≥50 years, enrolled in 2007–2008. Ophthalmic and anthropometric examinations were undertaken on all participants at baseline, and a questionnaire was administered, including medical and ophthalmic history. Participants were retraced in 2013–2014 for a second examination. Vital status was recorded for all participants through information from community members. Cumulative incidence of mortality, and its relationship with baseline VI and types of eye disease was estimated. Inverse probability weighting was used to adjust for non-participation.

**Primary outcome measures** Cumulative incidence of mortality in relation to VI level at baseline.

**Results** Of the baseline sample, 2170 (50%) were re-examined at follow-up and 407 (10%) were known to have died (adjusted risk of 11.9% over 6 years). Compared to those with normal vision (visual acuity (VA) ≥6/12, risk=9.7%), the 6-year mortality risk was higher among people with VI (<6/18 to ≥6/60; risk=28.3%; risk ratio (RR) 1.75, 95% CI 1.28 to 2.40) or severe VI (SVI)/blindness (<6/60; risk=34.9%; RR 1.98, 95% CI 1.04 to 3.80). These associations remained after adjustment for non-communicable disease (NCD) risk factors (mortality: RR 1.56, 95% CI 1.14 to 2.15; SVI/blind: RR 1.46, 95% CI 0.80 to 2.68). Mortality risk was also associated with presence of diabetic retinopathy at baseline (RR 3.18, 95% CI 1.98 to 5.09), cataract (RR 1.26, 95% CI 0.95 to 1.66) and presence of both cataract and VI (RR 1.57, 95% CI 1.24 to 1.98). Mortality risk was higher among people with age-related macular degeneration at baseline (with or without VI), compared with those without (RR 1.42, 95% CI 0.91 to 2.22 and RR 1.34, 95% CI 0.99 to 1.81, respectively).

**Conclusions** Visual acuity was related to 6-year mortality risk in this cohort of elderly Kenyan people, potentially

### Strengths and limitations of this study

► The cohort comprised a representative population-based sample in an area of ethnic, socioeconomic and educational diversity.
► There was a comprehensive assessment of ophthalmic characteristics and risk factors at baseline and follow-up.
► Data on mortality were collected through informant report, rather than from death certificates.
► There was a high loss to follow-up in this study, raising the possibility of selection bias.

because both VI and mortality are related to ageing and risk factors for NCD.

## INTRODUCTION

Visual impairment (VI) is common, affecting approximately 253 million people globally.[1] It can impact on different aspects of people's lives, including reducing quality of life, and increasing poverty and depression.[2–5] There is growing evidence from Europe, North America, Asia and Australia that VI and specific eye conditions are linked to increased risk of mortality,[6–17] but data are lacking for low-income and middle-income countries (LMICs), particularly from sub-Saharan Africa.

There are several potential pathways by which VI may be linked with mortality. Both VI and mortality are related to ageing, and so confounding or residual confounding may explain the reported associations. There are also common underlying risk factors for both VI and mortality, such as smoking, obesity and poverty. For instance, VI due to age-related macular degeneration (AMD) is more

**BMJ**

common among smokers,[18] and smoking increases risk of mortality. An underlying disease may also cause both VI and mortality, for instance, diabetic retinopathy (DR) is related to poor control of diabetes, which also causes increased mortality. People with VI may find it more difficult to seek healthcare, due to a range of barriers,[19] thereby increasing their mortality risk. Changes in the eye may be a marker of ageing or accelerated ageing,[20] and thereby linked to mortality. Finally, VI could acerbate frailty, depression and functional difficulties, all linked to increased mortality.[3 21 22]

It is important to explore whether there is an association between VI and mortality and, if there is, to identify possible pathways for this link, in order to understand how to reduce the vulnerability of people with VI to increased morbidity and mortality. Furthermore, these data may be useful to advocate for scaling up of eye health services in LMICs. The objective of this study was to investigate the association between VI and 6-year mortality risk within the Nakuru Eye Disease Cohort Study, a cohort of elderly Kenyan people.

## MATERIALS AND METHODS
The methodology of the Nakuru Eye Disease Cohort Study has been reported previously,[23] and is summarised here.

### Baseline study population
The baseline population-based survey was conducted in 2007/2008. A total of 100 clusters each of 50 participants were selected with a probability proportional to the size of the population across Nakuru district. Households were selected within clusters using a modified compact segment sampling method.[24] Eligible individuals were those aged ≥50 years living in the household for at least 3 months in the previous year, and multiple subjects could be included per household.

### Baseline ophthalmic and general examination
All participants were invited to undergo a comprehensive ophthalmic examination at a screening clinic.[23] The objectives of the survey and the examination process were explained to those eligible in the local dialect, in the presence of a witness. A subject was examined only after informed written (or thumbprint) consent was obtained.

All participants underwent Logarithm of the Minimum Angle of Resolution (logMAR) visual acuity (VA) testing on each eye separately and corrected VA (by refraction or pinhole) when less than 20/40 Snellen equivalent. Participants had two non-stereoscopic, digital, 45° fundus photographs (one disc and one macula centred) taken per eye by an ophthalmic clinical officer. Digital images were graded for the presence of AMD and DR at an approved grading centre (Moorfields Eye Hospital

**Table 1** Baseline characteristics of the Nakuru Eye Disease Cohort Study

| Baseline characteristics | | Deceased n=409 | Followed-up or known to be alive n=3032 | P value* | Unknown status n=973 | P value† |
|---|---|---|---|---|---|---|
| Age in years, mean (SD) | | 71.6 (12.8) | 62.4 (9.6) | <0.001 | 63.2 (10.6) | 0.60 |
| Sex, % (n) | Female | 42 (173) | 53 (1612) | <0.001 | 53 (516) | 0.59 |
| | Male | 58 (236) | 47 (1420) | | 47 (457) | |
| Tribe, % (n) | Kikuyu | 69 (283) | 62 (1881) | 0.02 | 61 (596) | 0.005 |
| | Kalenjin | 23 (93) | 24 (736) | | 19 (186) | |
| Education, % (n)‡ | None | 6 (26) | 9 (283) | <0.001 | 12 (114) | 0.21 |
| | Primary | 44 (179) | 32 (952) | | 33 (323) | |
| | Secondary | 43 (174) | 48 (1448) | | 44 (427) | |
| | Higher | 7 (29) | 11 (327) | | 11 (105) | |
| Residence, % (n) | Rural | 74 (303) | 71 (2167) | 0.43 | 51 (498) | <0.001 |
| | Urban | 26 (106) | 29 (865) | | 49 (475) | |
| SES quartile, % (n)‡ | Lower | 33 (136) | 24 (709) | 0.002 | 26 (247) | 0.17 |
| | Middle lower | 24 (96) | 26 (776) | | 23 (219) | |
| | Middle upper | 24 (97) | 26 (777) | | 23 (218) | |
| | Upper | 19 (79) | 24 (731) | | 29 (281) | |

*P value describes the strength of evidence that each variable is associated with mortality, among those where we know the mortality status (Null hypothesis is that the odds of death are equal in each category of the variable).
†P value describes the strength of evidence that each variable is associated with mortality status being missing (ie, comparison of known vs unknown mortality status). (Null hypothesis is that odds of knowing the mortality status of an individual at follow-up are equal in each group).
‡There were 27 missing values for education, and 48 missing values for SES.
SES, socioeconomic status.

| Weighted using inverse probability-weights | Overall | | Male | | Female | | <60 years | | ≥60 years | |
|---|---|---|---|---|---|---|---|---|---|---|
| | N | Risk per 1000/6 years (95% CI) | N | Risk per 1000/6 years (95% CI) | N | Risk per 1000/6 years (95% CI) | N | Risk per 1000/6 years (95% CI) | N | Risk per 1000/6 years (95% CI) |
| Visual acuity at baseline (better eye presenting) | | | | | | | | | | |
| All individuals | 409/3441 | 119 (106 134) | 236/1656 | 143 (123 165) | 173/1785 | 98 (84 113) | 86/1503 | 56 (45 70) | 323/1938 | 169 (151 188) |
| Normal (≥6/12) | 280/2901 | 97 (84 111) | 162/1378 | 118 (98 142) | 118/1523 | 77 (64 93) | 78/1420 | 54 (42 68) | 202/1481 | 138 (120 159) |
| Near normal (<6/12 to ≥6/18) | 27/170 | 158 (107 226) | 14/84 | 168 (104 258) | 13/86 | 148 (85 245) | 3/27 | 106 (32 301) | 24/143 | 168 (114 241) |
| VI (<6/18 to ≥6/60) | 77/275 | 283 (236 336) | 45/142 | 316 (252 388) | 32/133 | 249 (178 338) | 3/31 | 99 (31 271) | 74/244 | 307 (252 368) |
| Severe VI (<6/60 to ≥3/60) | 4/16 | 260 (86 566) | 2/10 | 187 (34 603) | 2/6 | 379 (51 875) | 2/2 | – | 2/14 | 136 (27 475) |
| Blind (<3/60) | 19/50 | 385 (245 548) | 13/30 | 436 (247 646) | 6/20 | 310 (133 570) | 0/6 | – | 19/44 | 438 (279 612) |
| Any VI (<6/18) | 100/341 | 297 (248 351) | 60/182 | 328 (268 395) | 40/159 | 262 (191 347) | 5/39 | 137 (57 297) | 95/302 | 318 (262 380) |

Table 2 The 6-year weighted mortality risk by level of visual impairment (VI) among the Nakuru Eye Disease Cohort Study Participants, stratified by age and gender

Reading Centre) by a senior grader, with adjudication by a clinician for confirmed cases and 5% of randomly selected images to ensure quality control. The presence of cataract was recorded by the ophthalmologist (WM) on slit-lamp examination after pupil dilation.

Detailed interviews were undertaken in the local language on demographic details, information on risk factors, socioeconomic status (SES) and full medical history.

A nurse performed and recorded measures of participants: height (Leicester Height Measure); weight (Seca 761) and three measures of blood pressure (Omron Digital Automatic Blood Pressure Monitor Model HEM907), each 10 min apart. Capillary blood was taken from all participants for random blood glucose (Accu-trend GC system).

### Assessment of vital status at follow-up
Follow-up was conducted from January 2013 to March 2014. A meeting was held approximately 1 week before the follow-up examination clinic for a given cluster. A list of study participants was given to the chief and a local village guide was recruited to assist location of the study participants. The village guide was someone who knew and was well known by the community (or the village chief him/herself). The advance team visited homes of baseline participants on the day prior to the examination clinic and confirmed their identity using the national identity cards and invited them to attend the examination clinic the following day. All identified participants were also asked to help locate baseline participants that had not been found.

Each local field guide was asked to classify the baseline study participant for that cluster as 'available', 'died', 'moved away' or 'unknown'. A participant was defined to have died if this was verified by at least two people from among the village chief, local guide or available study participant. Those who were known to have moved away were contacted when possible to either arrange follow-up at a more suitable location for the participant or to identify if they were alive or had died in the follow-up period. Any participant for whom nobody could identify as being alive or having moved away was recorded as 'unknown'.

### Definitions and statistical analyses
All participants who had complete examinations at baseline were considered 'at-risk' for mortality during follow-up. Follow-up status at 6 years was categorised as: (1) deceased (confirmed dead, as described above); (2) alive (ie, re-examined at follow-up, retraced but refused or unavailable at follow-up, or moved away but known to be alive) (3) unknown (ie, not retraced at follow-up, death not verified as described above or moved away but vital status unknown).

An SES score was developed based on information collected on job, housing conditions and ownership of material goods and livestock, based on previous work in the same population.[25] Hypertension was defined on the

**Table 3** Association between visual acuity (VA) category and 6-year mortality risk

| VA category | Age/sex-adjusted RR | Age/sex, SES* adjusted RR | Age/sex, NCD risk† factor adjusted RR | Fully adjusted ‡ RR |
|---|---|---|---|---|
| Normal (≥6/12) | Reference | Reference | Reference | Reference |
| Near normal (<6/12 to ≥6/18) | 0.92 (0.57–1.50) | 0.84 (0.51–1.39) | 0.87 (0.51–1.48) | 0.82 (0.48–1.41) |
| VI (<6/18 to ≥6/60) | 1.75 (1.28–2.40) | 1.77 (1.30–2.40) | 1.56 (1.13–2.16) | 1.56 (1.14–2.15) |
| SVI/blind (<6/60) | 1.98 (1.04–3.80) | 1.95 (1.01–3.76) | 1.51 (0.82–2.77) | 1.46 (0.80–2.68) |
| P value | 0.004 | 0.003 | 0.04 | 0.03 |

*SES=SES quartile, location, ethnic group, education.
†NCD risk factor=smoking, alcohol, diabetes, hypertension, BMI.
‡Age–sex, plus all SES and NCD risk factors.
BMI, body mass index; NCD, non-communicable disease; RR, risk ratio; SES, socioeconomic status; SVI, severe visual impairment; VI, visual impairment.

basis of the average of the second and third reading, with cut-offs used of systolic blood pressure ≥140 mm Hg and/or diastolic blood pressure ≥90 mm Hg and/or self-reported hypertension medication. Diabetes was defined as (1) self-reported in the history, or (2) random glucose of ≥11.0 mmol/L.

Statistical analysis was performed using STATA V.14 (Stata). All analyses accounted for the cluster survey design using Taylor linearised variance estimation to calculate standard errors. Pearson $X^2$ tests corrected for the survey design were used to calculate p values to assess differences between participants whose mortality status is known and those where mortality status is unknown, that is, lost to follow-up (LTFU).

An inverse probability-weighting (IPW) model[26] was developed, in order to allow estimation of mortality risk while accounting for those LTFU. Multivariable logistic regression was used to identify independent baseline covariates associated with LTFU. Covariates for which there was the evidence of univariable association with the outcome (p<0.10 across all categories of the variable) were kept in a multivariable model (age, sex, rural/urban and mother tongue). From this final model, the probability of being followed up was estimated, based on the presence or absence of each of these baseline covariates. The inverse of this probability formed the weighting to be applied to account for those LTFU.

The final step was to remove those individuals LTFU from the cohort, so that all subsequent analysis would be performed on only those with complete outcome records, with IPW applied to account for those LTFU. A sensitivity analyses for this approach involved repeating the analyses without applying IPW (ie, standard unweighted complete case analysis), and assessing the impact on the results.

Six-year mortality risk was calculated by dividing the number of deaths identified at follow-up by the number of people at risk at baseline. The 95% CIs were estimated assuming a Poisson distribution of events. This was done for the population overall, and stratified by each covariate.

Age/sex-adjusted risk ratios (RRs) for each covariate in relation to mortality were estimated using a Poisson regression model with robust error variance to allow for the clustered design and including IPW. Mortality status was the binary outcome and the distribution was assumed to be Poisson. These analyses were adjusted for the clustered design, as well as the use of IPW, by setting the clusters as the primary sampling unit and weighting using the inverse probability of being followed up. The model was further adjusted using a set of four SES variables only (SES quartile, location, ethnic group, education), then a set of five non-communicable disease (NCD) risk factors only (smoking, alcohol, diabetes, hypertension, body mass index (BMI)) before finally adjusting for all nine SES and NCD variables.

### Patient and public involvement
Patients and the public were not directly involvement in the development of the research question and outcome measures, or the design or conduct of study. There are no plans to disseminate results directly to the study participants.

### RESULTS
At baseline, 4414 participants were examined. The follow-up assessment was conducted, on average, 5.6 years (SD 0.6) after the baseline, expressed for simplicity as 6 years (meaning that there were 6 years between the baseline and follow-up wave, rather than that each participant was followed up on average for 6 years as the time of LTFU or death was not known for individuals). Of the baseline participants, 3032 were known to be alive at follow-up (69% 2170 re-examined at the follow-up plus 862 known to be alive but not re-examined), 409 (9%) were known to have died and 973 had unknown vital status (22%).

Table 1 provides the baseline characteristics of participants who had died during the follow-up, those were re-examined at follow-up and those who were LTFU. In comparison to those who had died, those who were re-examined were younger, more likely to be female, Kalenjin speakers, and had higher SES, while those of unknown

status were more likely to be of 'other' tribes and urban residence.

Table 2 shows the weighted 6-year mortality risk by level of VI. Overall, the 6-year mortality risk was 11.9% over 6 years. Risk increased with worsening levels of VI, from 9.7% (95% CI 8.4% to 11.1%) among those with normal vision to 38.5% (95% CI 24.5% to 54.8%) among those who were blind. This pattern was observed in both males and females, but was less clear in people aged <60 years given the low mortality in this group and consequent small numbers. In each subgroup, the lowest risk of mortality was among people with normal vision. Mortality risk among people with VI was higher for males than for females, and among those ≥60 years vs <60 years. Estimates changed little after weighting for LTFU (online supplementary table for unweighted estimates).

Compared with those with normal vision (VA >6/12, risk=9.7%), the mortality risk was significantly higher among people with VI (VA <6/18 to ≥6/60; risk=28.3%; RR 1.75, 95% CI 1.28 to 2.40) or severe VI (SVI)/blindness (VA <6/60: risk=34.9%; RR 1.98, 95% CI 1.04 to 3.80) (table 3). There was a weakening of the association after adjustment for NCD risk factors or full adjustment for both SES and NCD risk factors, although the overall trends between worsening vision and increased 6-year mortality risk remained evident (VI: RR 1.56, 95% CI 1.14 to 2.15; SVI/blind: RR 1.46, 95% CI 0.80 to 2.68).

People with any VI had a higher mortality risk than those without VI (29.7% vs 9.7%; RR 1.54, 95% CI 1.22 to 1.93), and this association remained after adjustment for SES and NCD risk factors (RR 1.37, 95% CI 1.10 to 1.71) (table 4). Other risk factors associated with 6-year mortality risk after comprehensive adjustment included increasing age (oldest vs youngest age group: RR 4.68, 95% CI 3.55 to 6.18) and diabetes (RR 2.34, 95% CI 1.81 to 3.03). Being underweight was associated with an increased 6-year mortality risk (underweight vs normal: RR 1.60, 95% CI 1.24 to 2.07).

Risk of mortality was analysed by prevalence of specific eye diseases at baseline (table 5). The presence of cataract (or aphakia/pseudophakia) and any VI (ie, VA <6/18 in better eye) was associated with higher mortality risk (RR 1.57, 95% CI 1.24 to 1.98), whereas cataract alone (or aphakia/pseudophakia) was not (RR 1.26, 95% CI 0.95 to 1.66). Mortality risk was higher among people with AMD at baseline (with or without VI), compared with those without, although these associations were not statistically significant (RR 1.42, 95% CI 0.91 to 2.22 and RR 1.34, 95% CI 0.99 to 1.81, respectively). DR was associated with a threefold increased mortality risk (RR 3.18, 95% CI 1.98 to 5.09). The number of people with DR and any VI was too small to make meaningful inferences.

## DISCUSSION

VI was associated with increased mortality risk during 6 years of follow-up in a cohort of elderly Kenyan people. The risk of mortality increased with worsening vision. This association was reduced after adjustment for the presence of NCD risk factors, and to a lesser extent for SES indicators. Among eye conditions, DR was most strongly associated with mortality risk, although the number affected was small. Cataract with VI was also associated with elevated mortality, as were AMD and cataract without visual loss at baseline (although these estimates lacked precision).

Previous studies have also shown a positive relationship between VI and mortality, with evidence available from the USA,[6–9] UK,[10] Australia,[11 12] Japan,[13] Singapore,[14 15] China[16] and India.[17] Others have failed to find evidence for this association, including in India,[27] Iceland[28] and Taiwan.[29] Data from LMICs are sparse, in particular for sub-Saharan Africa, and so comparison of our study findings to those from similar settings is not possible.

On the basis of our findings, and those in the wider literature, consideration can be given to the potential pathway for the association between VI and mortality. There was clear evidence for confounding by age, as both VI and mortality are independently related to older age. Consequently, imperfect adjustment for age may have allowed for residual confounding as a partial explanation for the association. There was little evidence for confounding by SES, although in this setting high SES was associated with greater prevalence of NCD risk factors,[30] and a somewhat reduced mortality risk. The presence of NCD risk factors may also act as confounders of the association of VI on mortality, since the association was attenuated after adjustment for these indicators, as found in other studies.[10] Significant associations persisted, however, between VI and mortality after comprehensive multivariable adjustment in this study, as occurred in previous studies,[12–14 17 31] suggesting that residual confounding or direct effects of VI on mortality may be operational.

Exploring the relationship between different eye conditions and mortality may help to clarify whether independent biological pathways exist. DR is known to be associated with increased mortality,[15 32] as was also shown in this study. This link is unsurprising given the well-known relationship between uncontrolled diabetes with both DR and mortality. However, the relatively small number of people with DR in this population means that this link cannot be the sole driver of the VI-mortality association. Our study, as well as others, has shown cataract to be associated with increased mortality,[6 12 17 33 34] though this association is not always demonstrated.[15 35] Some studies have suggested that this relationship varies by cataract type.[8 34 36–38] It is hypothesised that the association between cataract and mortality arises as lens opacification (cataract) is an indicator of accelerated ageing.[20 39] The evidence for a link between AMD and mortality is more complex; some studies show that late AMD is associated with mortality, but not early AMD.[6 40–42] Others found no association between AMD and mortality,[15 33 35] or only among women.[43]

There are other potential pathways between VI and mortality not explored in this study. For example, NCD risk factors may be mediators of the effect of VI on

**Table 4** Multivariable analysis of baseline covariables and 6-year mortality risk in the Nakuru Eye Disease Cohort Study

| | No at risk | Deaths | Risk per 1000/6 years (95% CI) | Age/sex-adjusted risk ratio | Age/sex/ socioeconomic status (SES) adjusted risk ratio | Age/sex/SES- non-communicable disease (NCD) risk factor adjusted risk ratio |
|---|---|---|---|---|---|---|
| Any visual impairment (<6/18) | | | | | | |
| No | 3071 | 307 | 100 (87 to 115) | Reference | Reference | Reference |
| Yes | 341 | 100 | 297 (248 to 351) | 1.54 (1.22 to 1.93) | 1.55 (1.24 to 1.94) | 1.37 (1.10 to 1.71) |
| Gender | | | | | | |
| Male | 1656 | 236 | 143 (123 to 165) | Reference | Reference | Reference |
| Female | 1785 | 173 | 98 (84 to 113) | 0.74 (0.63 to 0.87) | 0.68 (0.56 to 0.83) | 0.82 (0.63 to 1.06) |
| Age | | | | | | |
| 50–59 | 1503 | 86 | 56 (45 to 70) | Reference | Reference | Reference |
| 60–69 | 1036 | 96 | 94 (77 to 115) | 1.64 (1.24 to 2.17) | 1.58 (1.19 to 2.09) | 1.40 (1.07 to 1.83) |
| 70–79 | 571 | 107 | 191 (161 to 224) | 3.27 (2.55 to 4.20) | 3.15 (2.39 to 4.15) | 2.74 (2.09 to 3.60) |
| 80+ | 331 | 120 | 363 (311 to 419) | 6.36 (4.85 to 8.33) | 5.76 (4.39 to 7.57) | 4.68 (3.55 to 6.18) |
| SES risk factors | | | | | | |
| Location | | | | | | |
| Rural | 2470 | 303 | 123 (108 to 140) | Reference | Reference | Reference |
| Urban | 971 | 106 | 110 (83 to 145) | 1.14 (0.90 to 1.45) | 1.15 (0.90 to1.48) | 1.18 (0.91 to 1.53) |
| SES quartile | | | | | | |
| Lower | 845 | 136 | 164 (137 to 197) | Reference | Reference | Reference |
| Lower middle | 872 | 96 | 110 (91 to 134) | 0.75 (0.60 to 0.93) | 0.72 (0.58to 0.91) | 0.75 (0.59 to 0.95) |
| Upper middle | 874 | 97 | 110 (89 to 134) | 0.86 (0.66 to 1.11) | 0.82 (0.63 to 1.06) | 0.84 (0.65 to 1.09) |
| Upper | 810 | 79 | 99 (78 to 126) | 0.89 (0.69 to 1.14) | 0.78 (0.58 to 1.05) | 0.76 (0.56 to 1.03) |
| Ethnic group | | | | | | |
| Kikuyu | 2164 | 283 | 131 (116 to 148) | Reference | Reference | Reference |
| Kalenjin | 829 | 93 | 113 (91 to 139) | 0.83 (0.66 to 1.03) | 0.78 (0.62 to 0.99) | 0.81 (0.63 to 1.04) |
| Other | 448 | 33 | 76 (48 to 119) | 0.84 (0.56 to 1.26) | 0.82 (0.55 to 1.21) | 0.81 (0.55 to 1.20) |
| Education | | | | | | |
| No education | 309 | 26 | 86 (58 to 125) | Reference | Reference | Reference |
| Primary | 1131 | 179 | 161 (138 to 188) | 0.95 (0.65 to 1.41) | 0.96 (0.63 to 1.46) | 0.97 (0.63 to 1.50) |
| Secondary | 1622 | 174 | 107 (91 to 125) | 0.85 (0.59 to 1.22) | 0.84 (0.57,1.24) | 0.87 (0.59 to 1.28) |
| College/Uni | 356 | 29 | 81 (52 to 124) | 0.94 (0.57 to 1.55) | 0.95 (0.57 to 1.55) | 1.01 (0.62 to 1.64) |
| Risk factors for NCD | | | | | | |
| Smoking | | | | | | |

**Table 4** Continued

| | No at risk | Deaths | Risk per 1000/6 years (95% CI) | Age/sex-adjusted risk ratio | Age/sex/ socioeconomic status (SES) adjusted risk ratio | Age/sex/SES- non-communicable disease (NCD) risk factor adjusted risk ratio |
|---|---|---|---|---|---|---|
| Never | 2396 | 256 | 107 (92 to 124) | Reference | Reference | Reference |
| Former | 252 | 33 | 131 (93 to 180) | 1.20 (0.85 to 1.69) | 1.20 (0.85 to 1.71) | 1.15 (0.79 to 1.66) |
| Current | 775 | 120 | 156 (132 to 184) | 1.19 (0.93 to 1.52) | 1.17 (0.91 to 1.50) | 1.19 (0.91 to 1.55) |
| Diabetes | | | | | | |
| No | 3202 | 354 | 111 (98 to 125) | Reference | Reference | Reference |
| Yes | 216 | 54 | 248 (194 to 313) | 2.16 (1.69 to 2.77) | 2.20 (1.69 to 2.84) | 2.34 (1.81 to 3.03) |
| Hypertension | | | | | | |
| No | 1670 | 179 | 108 (92 to 126) | Reference | Reference | Reference |
| Yes | 1737 | 229 | 132 (113 to 53) | 1.08 (0.90 to 1.29) | 1.06 (0.88 to 1.28) | 1.11 (0.92 to 1.34) |
| Alcohol | | | | | | |
| Never | 1335 | 117 | 87 (73 to 104) | Reference | Reference | Reference |
| Former | 1520 | 221 | 147 (125 to 171) | 1.18 (0.95 to 1.48) | 1.18 (0.94 to 1.49) | 1.15 (0.90 to 1.46) |
| Current | 559 | 70 | 125 (100 to 157) | 1.15 (0.86 to 1.53) | 1.18 (0.87 to 1.60) | 1.08 (0.77 to 1.51) |
| Body mass index | | | | | | |
| Underweight | 468 | 99 | 216 (169 to 271) | 1.55 (1.20 to 2.00) | 1.57 (1.22 to 2.02) | 1.60 (1.24 to 2.07) |
| Normal | 1697 | 199 | 117 (101 to 136) | Reference | Reference | Reference |
| Overweight | 779 | 66 | 86 (66 to 111) | 0.90 (0.69 to 1.18) | 0.87 (0.66 to 1.14) | 0.81 (0.62 to 1.05) |
| Obese | 447 | 32 | 73 (52 to 100) | 0.89 (0.63 to 1.25) | 0.86 (0.59 to 1.24) | 0.83 (0.57 to 1.20) |

mortality (rather than confounders) for reasons such as lower ability to access NCD treatment, less exercise, poorer diets and so on. Consequently, the association between VI and mortality adjusted by NCD risk factors would be an underestimate of the total effect. There were also concerns about the accuracy of assessment of visual fields in this population. Consequently, it was not possible to determine the presence of glaucoma at baseline, although others have suggested a link between glaucoma and mortality.[15 44] We also did not assess the impact of VI in accessing healthcare, although the Australian Blue mountain study showed that difficulties in walking explained some of the link between VI and mortality.[11] Only 18 people with cataract underwent cataract surgery during the follow-up period, so it was not possible to assess the impact on mortality.

There are several further limitations of the study, which need to be considered when interpreting the findings. There was a lack of data on date of death, and no verification from death certificates, as these are rarely available in many African settings,[45] including Kenya. Cause of death could not be determined, and so we could not assess whether the relationship was stronger between VI and specific causes of mortality, notably cardiovascular and non-cancer causes, as demonstrated in previous studies,[15 46–48] which would lend weight to a biological pathway for the association. The follow-up study was conducted after a period of postelection violence in the area. Consequently, there was a high LTFU in this study, raising the possibility of selection bias influencing the findings, although patterns changed little after weighting for LTFU. We did not adjust for the population sampling weights in our analysis, and so there could be concerns about the representativeness of the sample, although the selected sample had a similar demographic distribution to the general population.[49] Furthermore, the mortality rate may have been higher in this period, due to violence, and may have biased the association with VI if these deaths were disproportionally among people with VI, or among younger people (with lower prevalence of VI). Date of

**Table 5**  Risk of mortality during 6 years of follow-up by the presence of specific eye diseases at baseline

| | No at risk | Deaths | Risk per 1000/6 years (95% CI) | Age/sex- adjusted risk ratio |
|---|---|---|---|---|
| **Cataract present** | | | | |
| No | 1921 | 142 | 72.5 (59.1 to 88.6) | Reference |
| Yes | 1478 | 265 | 181.3 (161.6 to 202.8) | 1.26 (0.95 to 1.66) |
| **Cataract and visual impairment (VI) present (<6/18)** | | | | |
| No | 3103 | 313 | 100.7 (87.5 to 115.5) | Reference |
| Yes | 296 | 94 | 322 (267.3 to 381.8) | 1.57 (1.24 to 1.98) |
| **Age-related macular degeneration (AMD) present** | | | | |
| No | 2270 | 225 | 99.2 (83.3 to 117.8) | Reference |
| Yes | 319 | 57 | 183.4 (141.3 to 234.5) | 1.34 (0.99 to 1.81) |
| **AMD and VI present (<6/18)** | | | | |
| No | 2529 | 265 | 105.5 (89.9 to 123.4) | Reference |
| Yes | 60 | 17 | 282.7 (178.5 to 416.8) | 1.42 (0.91 to 2.22) |
| **Diabetic retinopathy (DR) present** | | | | |
| No | 2513 | 264 | 105.7 (90.4 to 123.3) | Reference |
| Yes | 55 | 18 | 318.4 (199.4 to 467.0) | 3.18 (1.98 to 5.09) |
| **DR and VI present (<6/18)** | | | | |
| No | 2563 | 280 | 109.9 (94.1 to 128.0) | Reference |
| Yes | 5 | 2 | 401.6 (37.5 to 920.4) | 2.54 (0.57 to 11.36) |

LTFU or death was not recorded, and so survival analysis was not possible. Another concern is that reports of local informants were used to categorise some people who had moved away as 'known to be alive', which may have created inaccuracies. The study may have been underpowered for some of the subgroup analyses, such as assessing the link between type of eye disease and mortality. We did not evaluate the association of different subtypes of AMD or cataract in relation to mortality because of small numbers. VI classification did not include loss of visual fields, and so the prevalence of functionally significant sight loss may have been underestimated. Self-reported diabetes was not confirmed (eg, from medical records). In terms of strengths, this was the first study of its kind in sub-Saharan Africa to assess the association between VI and mortality. The study participants comprised a representative population-based sample in an area of ethnic, SES and educational diversity. There was a comprehensive assessment of ophthalmic characteristics and risk factors at baseline, and every attempt was made to follow up all participants, and to record vital status.

In conclusion, VA was related to 6-year mortality risk in this cohort of elderly Kenyan people. The most likely explanation for the association is that both VI and mortality are related to ageing and NCD risk factors. The implication is that continuity of care is needed, as people with VI require linkages to preventative and treatment services. Furthermore, we must advocate for the scale-up of eye care services in Kenya, as VI is linked to premature mortality.

**Author affiliations**
[1]International Centre for Eye Health, Clinical Research Department, London School of Hygiene and Tropical Medicine, London, UK
[2]International Centre for Evidence in Disability, Clinical Research Department, London School of Hygiene and Tropical Medicine, London, UK
[3]Rwanda International Institute of Ophthalmology and Dr. Agarwal's Eye Hospital, Kigali, Rwanda
[4]MRC Tropical Epidemiology Group, Department of Infectious Disease Epidemiology, London School of Hygiene & Tropical Medicine, London, UK
[5]Ministry of Health, Nairobi, Kenya
[6]Kitale Eye Unit, Ministry of Health Trans Nzoia County, Kitale, Kenya
[7]Department of Non-Communicable Disease Epidemiology, London School of Hygiene & Tropical Medicine, London, UK

**Contributors**  HK had full access to all the data in the study and takes responsibility for the integrity of the data and the accuracy of the data analysis. Study concept and design: AB, WM, AF, MB and HK. Acquisition, analysis or interpretation of data: AB, KW, HR, HAW, DM and MB. Drafting of the manuscript: HK, AB and DM. Critical revision of the manuscript for important intellectual content: AB, WM, MG, KW, HR, HAW, AF, MB and HK. Statistical analysis: HK, AB, KW, HAW and

DM. Obtained funding: AB and HK. Administrative, technical or material support: AB, WM, MG, HR and HK. Study supervision: AB, AF, MB and HK.

**Funding** This study was supported by grant G1001934 from the Medical Research Council, grant 1310 from Fight for Sight, the British Council for the Prevention of Blindness, and the International Glaucoma Association (Dr Bastawrous). MB is supported by grant 098481/Z/12/Z from the Wellcome Trust. HAW is supported by grant G0700837 from the Medical Research Council and Department for International Development.

**Disclaimer** The funding sources had no role in the design and conduct of the study; collection, management, analysis and interpretation of the data; preparation, review or approval of the manuscript; and the decision to submit the manuscript for publication.

**Competing interests** None declared.

**Patient consent for publication** Obtained.

**Ethics approval** The study adhered to the tenets of the Declaration of Helsinki and was approved by the Ethics Committee of London School of Hygiene and Tropical Medicine at both baseline and follow-up (LSHTM Ref 6192). Baseline approval was provided by the Kenya Medical Research Institute Ethics Committee and by the African Medical and Research Foundation (AMREF) Ethics Committee, Kenya for the follow-up (AMREF-ESRC P44/12). For both phases, approval was granted by the Rift Valley Provincial Medical Officer and the Nakuru District Medical Officer of Health. Approval was sought from the administrative heads in each cluster.

**Provenance and peer review** Not commissioned; externally peer reviewed.

**Data sharing statement** Data are available on request from AB or HK.

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
