## [Reviewer comments · BMJ Open]

ARTICLE DETAILS

TITLE (PROVISIONAL)	Mortality during six years of follow-up in relation to visual impairment and eye disease: Results from a population-based cohort study of people aged 50 years and above in Nakuru, Kenya
AUTHORS	Kuper, Hannah; Mathenge, Wanjiku; Macleod, David; Foster, Allen; Gichangi, Michael; Rono, Hillary; Wing, Kevin; Weiss, Helen; Burton, Matthew J; Bastawrous, Andrew

VERSION 1 – REVIEW

REVIEWER	Fei Yu UCLA, USA
REVIEW RETURNED	25-Feb-2019

GENERAL COMMENTS	Six-year mortality risk in relation to visual impairment and eye disease: Results from a population-based cohort study of people aged 50 years and above in Nakuru, Kenya The authors have clarified most issues raised in previous reviews. There are a few remaining concerns It seems that the authors corrected for the cluster survey design and IPW weights in the analyses, but not the original population sampling weights from the baseline. Otherwise, it is not clear how the authors can account for both weights in one analysis. It might be a tough choice between the population sampling weights (for a representative sample) and IPW (for potential selection bias), and the authors need to aware of this limitation and its impact on the study results. Also, as this might be too technical for the authors to put in the manuscript, it might be helpful for the authors to describe this in more details in an appendix so that readers will be able to understand and replicate such analysis. The IPW model only included four variables, so that the probability of LTFU might not be estimated precisely. The authors might consider to examine more baseline factors to improve the estimate of the probability of LTFU. Otherwise, I doubt the usefulness of IPW model, and this might explain that no differences were observed between weighted unweighted results. I would suggest the authors to change the title as “mortality during six years of follow-up” instead of “six-year mortality”, which generally means the mortality at six year, such as the estimate obtained from a Kaplan-Meier analysis. The authors’ results were clearly an overestimate of a “six-year mortality”. It would be fine for the authors to use Poisson models to obtain risk ratio estimates more easily, but the authors should also be
---

	aware the assumption of a binary variable following a Poisson distribution, instead of a Bernoulli distribution used in logistic regression models, which is a more reasonable assumption for a binary variable. Table 3: I assume that the fully adjusted model included age and sex in the model, and this should be clarified in the column heading or table footnote.
--	---

VERSION 1 – AUTHOR RESPONSE

Reviewer: 1

It seems that the authors corrected for the cluster survey design and IPW weights in the analyses, but not the original population sampling weights from the baseline. Otherwise, it is not clear how the authors can account for both weights in one analysis. It might be a tough choice between the population sampling weights (for a representative sample) and IPW (for potential selection bias), and the authors need to aware of this limitation and its impact on the study results. Also, as this might be too technical for the authors to put in the manuscript, it might be helpful for the authors to describe this in more details in an appendix so that readers will be able to understand and replicate such analysis.

Response: The selected sample was previously shown to be relatively representative of the general population (Mathenge W, Bastawrous A, Foster A, Kuper H. The Nakuru posterior segment eye disease study: methods and prevalence of blindness and visual impairment in Nakuru, Kenya. *Ophthalmology*. 2012 Oct;119(10):2033-9.). We therefore adjusted for IPW, but not population sampling weights. We have specified in the limitations that we did not adjust for population sampling weights in the analyses, but that we believed that the sample was representative.

The IPW model only included four variables, so that the probability of LTFU might not be estimated precisely. The authors might consider to examine more baseline factors to improve the estimate of the probability of LTFU. Otherwise, I doubt the usefulness of IPW model, and this might explain that no differences were observed between weighted unweighted results.

Response: Covariates for which there was evidence of univariable association with the outcome ($p < 0.10$ across all categories of the variable) were kept in a multivariable model (age, sex, rural/urban and mother tongue). We do not believe that including further variables unrelated to the outcome would improve the estimate of the probability of LTFU.

I would suggest the authors to change the title as “mortality during six years of follow-up” instead of “six-year mortality”, which generally means the mortality at six year, such as the estimate obtained from a Kaplan-Meier analysis. The authors’ results were clearly an overestimate of a “six-year mortality”.

Response: We have made this change.

It would be fine for the authors to use Poisson models to obtain risk ratio estimates more easily, but the authors should also be aware the assumption of a binary variable following a Poisson distribution,

instead of a Bernoulli distribution used in logistic regression models, which is a more reasonable assumption for a binary variable.

Response: We thank the reviewer for reminding us to consider the assumptions required in order for the Poisson distribution to approximate the binomial distribution. We think in this instance that using a Poisson model was appropriate.

Table 3: I assume that the fully adjusted model included age and sex in the model, and this should be clarified in the column heading or table footnote.

Response: This clarification has been made.